# Acupuncture for Poor Ovarian Response: A Randomized Controlled Trial

**DOI:** 10.3390/jcm10102182

**Published:** 2021-05-18

**Authors:** Jihyun Kim, Hoyoung Lee, Tae-Young Choi, Joong Il Kim, Byoung-Kab Kang, Myeong Soo Lee, Jong Kil Joo, Kyu Sup Lee, Sooseong You

**Affiliations:** 1Clinical Medicine Division, Korea Institute of Oriental Medicine, Daejeon 34054, Korea; kimjh763@kiom.re.kr (J.K.); lhoyoung@nate.com (H.L.); superoung@kiom.re.kr (T.-Y.C.); jikim@kiom.re.kr (J.I.K.); mslee@kiom.re.kr (M.S.L.); 2Department of Obstetrics and Gynecology, Pusan National University, Busan 49241, Korea; bkkang@kiom.re.kr (B.-K.K.); jkjoo@pusan.ac.kr (J.K.J.)

**Keywords:** acupuncture, network analysis, poor ovarian response, oocyte retrieval, in vitro fertilization

## Abstract

Acupuncture is believed to improve ovarian reserve and reproductive outcomes in women undergoing in vitro fertilization (IVF). This study was conducted to evaluate the effect of network-optimized acupuncture followed by IVF on the oocyte yield in women showing a poor ovarian response. This study was an exploratory randomized controlled trial conducted from June 2017 to January 2020 at the Pusan National University Hospital. Women diagnosed with poor ovarian response were enrolled and randomly divided into two groups: IVF alone and Ac + IVF groups (16 acupuncture sessions before IVF treatment). Eight acupoints with high degree centrality and betweenness centrality were selected using network analysis. Among the participants, compared with the IVF treatment alone, the acupuncture + IVF treatment significantly increased the number of retrieved mature oocytes in women aged more than 37 years and in those undergoing more than one controlled ovarian hyperstimulation cycle. The negative correlation between the number of retrieved mature oocytes and consecutive controlled ovarian hyperstimulation cycles was not observed in the Ac + IVF group irrespective of the maternal age. These findings suggest that physicians can consider acupuncture for the treatment of women with poor ovarian response and aged > 37 years or undergoing multiple IVF cycles.

## 1. Introduction

Acupuncture, a system of complementary Oriental medicine involving needle insertion into specific points of the body called acupoints, is a clinically proven therapy for the treatment of numerous conditions, including persistent back pain, stroke, and allergenic arthritis [1]. Acupuncture is also considered in the intervention for the improvement of women’s reproductive health and provides effective treatment for women’s reproductive disorders, including dysmenorrhea, abnormal menstrual flow, premenstrual symptoms, and irregular menstrual cycles, which has become a topic of interest, based on the findings of several clinical trials [2,3,4]. Acupuncture is thought to improve women’s reproductive disorders by modulating endogenous physiological mechanisms, including modulation of neuroendocrine factors and cytokine levels, increasing blood flow to the ovaries and uterus, and relieving stress, anxiety, and depression [5,6]. However, the major complications associated with acupuncture are bacterial infections, tissue injury, edema, and scarring at the needle insertion sites [7]. Guidelines for acupuncture should be strictly implemented to avoid such complications. Although acupuncture is popular in many countries, the optimal acupoints and their mechanisms of action have not been sufficiently investigated. Recently, a systematic framework for the acupoint combination network was used to identify acupoints that are frequently used together to treat body pain and insomnia [8]. Recently, numerous clinical trials and systematic reviews have consistently attempted to establish a medical standard for acupuncture and determine the effectiveness of acupuncture for the treatment of female reproductive disorders [9].

Infertility due to poor ovarian response (POR) is associated with poor reproductive prognosis among patients undergoing assisted reproductive technology treatment owing to a decrease in the ovarian follicular pool [10]. In such cases, maximizing the oocyte yield can lead to higher cumulative birth rates. However, optimal treatment options should be considered [11]. In contrast, some clinical trials showed that needling women with diminished ovarian reserves at multiple acupoints could yield positive outcomes [12]. However, to the best of our knowledge, no previous report has used systematic network analysis to determine effective acupoints for the treatment of women with POR. Therefore, this study aimed to perform a systematic network analysis to determine the optimal combination of acupoints for the treatment of women with POR and to investigate the effect on oocyte yields of acupuncture followed by ovulation induction in women with POR undergoing in vitro fertilization (IVF). Because of the deterioration of ovarian response occurs across repeated controlled ovarian hyperstimulation (COH) cycles or based on maternal age [13,14], the correlation of the number of retrieved oocytes, assisted reproductive technology cycles, and maternal age with pre-acupuncture treatment was additionally analyzed.

## 2. Materials and Methods

### 2.1. Extraction of Candidate Acupoints for POR Treatment

Relevant articles were retrieved from PubMed; Cochrane Library; seven Korean medical databases (the Korea Studies Information, DBPIA, Oriental Medicine Advanced Searching Integrated System, Research Information Service System, KoreaMed, Town Society of Science Technology, and Korean National Assembly Library), and one Chinese medical database, namely, the China National Knowledge Infrastructure for articles published from January 2000 to January 2017. The employed search terms were acupuncture, electro-acupuncture, acupuncture points, needling, transcutaneous electric nerve stimulation, poor ovarian reserve, diminished ovarian reserve, POR in Korean, Chinese, and English.

Two authors independently screened the retrieved articles by reading the title and abstract to identify potentially eligible clinical studies. After initial screening, a more thorough investigation was performed on reading the full text. All types of studies using acupuncture for POR treatment were included. We excluded trials in which acupuncture was part of a complex intervention. Reports that did not provide details of the acupoints used were also excluded. Data on study-level characteristics, treatment, outcomes, results, and acupoints were extracted. Any disagreements between two authors were resolved by discussion or, where necessary, arbitrated by a third author.

### 2.2. Selection of Optimal Acupoint Combination Using Network Analysis

Acupoint data were obtained from 13 clinical studies. To determine the suitable acupoint combination for POR treatment, patterns of occurrence and associations among acupoints were analyzed using network analysis (Figure 1a). The acupoint combination network was constructed based on the proportion of co-occurrence among acupoints (Figure 1b, upper triangle). The network structure was simplified, and the network backbone was extracted without loss of generality using minimum spanning tree analysis based on Kruskal’s algorithm (Figure 1b, lower triangle) [15,16]. To characterize the segregation of the minimum spanning tree network, the modularity was computed using the Louvain community detection algorithm [17,18]. Centrality measures such as node degree and betweenness were computed to identify topologically important acupoints, commonly referred to as hubs in the minimum spanning tree network.

### 2.3. Study Design

The clinical trial was conducted at the infertility center of Pusan National University (Busan, Republic of Korea, protocol version: KI-17-D-001 Version 2.0) from June 2017 to January 2020. This trial enrolled women diagnosed with POR at the infertility center of Pusan National University Hospital (Busan, Korea), according to the Bologna criteria published in 2011 by the European Society of Human Reproduction and Embryology [19] and included women with (1) age > 40 years or any other risk factor for POR; (2) a history of POR (less than three oocytes retrieved using a conventional stimulation protocol); and (3) an abnormal ovarian reserve test finding (total antral follicle count < 5–7 or Anti-Müllerian hormone < 0.5–1.1 ng/mL). Participants were included if they met at least two of these criteria. Two episodes of POR after the stimulation of ovulation were sufficient for the diagnosis of POR. Patients enrolled in this study were required to provide informed consent by signing written forms.

The randomization was implemented with the aid of sealed envelopes containing random assignment codes. The codes were generated using the block randomization method by a statistician, who was unaware of the aim of the study, using SAS, version 9.4 (SAS Institute Inc., Cary, NC, USA) [20]. Eligible patients were randomly assigned in a 1:1 ratio into either of the two trial groups: the IVF treatment group (IVF group) or the combined acupuncture and IVF treatment group (Ac + IVF group). The study was designed according to a previous study that focused on improving ovarian function in patients with POR (Figure 2) [21].

Participants in the Ac + IVF group were required to undergo 16 sessions of acupuncture with a 2-day (+1 day) interval between sessions with needling at eight selected acupoints (CV3, CV4, EX-CA1, SP6, KI3, SP10, ST36, and LR3) and abdominal irradiation with infrared rays for 30 min approximately 2 months before the induction of ovulation for oocyte retrieval. Participants in the IVF group were not scheduled to visit the clinic until the next session for ovulation induction. Protocols for the induction of ovulation in each patient were based on minimal stimulation [22], and the dosage and frequency of hormonal drugs were determined at the physician’s discretion based on the patient’s response to drugs.

The detailed study protocol has been presented in our previous study [21] and was approved by the Institutional Review Board of Pusan National University (IRB No. D-1706-017-056). The protocol used in this study was registered at the Clinical Research Information Service (CRIS, registration number: KCT0002623) in Korea, which is one of the primary registries of the WHO International Clinical Trials Registry Platforms.

The sample size was calculated using a bilateral significance of 5%, a dispensing ratio of 1:1, and a dropout rate of 10% as the power of the study [21]. Owing to the absence of related studies, the sample size comprised 35 patients in each group based on the results from the study by Zheng et al. [23]. However, the trial was performed with a smaller number of patients than that comprising the calculated sample size owing to valid IRB approval. Although the target number of patients could not be enrolled by the end of the clinical trial period, the goal of an exploratory study was achieved.

### 2.4. Safety and Effectiveness

Clinical laboratory tests (i.e., hematology, blood chemistry, and urinalysis) and endocrine function tests were performed at the time of screening and after acupuncture treatment in Pusan National University hospital by routine methods. Serum follicle-stimulating hormone (FSH), luteinizing hormone (LH), estradiol (E2), and dehydroisoandrosterone sulfate (DHEA-S) were detected using a radioimmunoassay. Serum thyroid-stimulating hormone (TSH) and prolactin (PRL) were detected using an immunoradiometric assay. Serum AMH was detected using an enzyme immunoassay. For FSH, LH, E2, TSH, and PRL, the inter- and intra-assay coefficients of variations (CV) were <5.72% sensitivity, 0.09 mIU/mL), <8.24% (sensitivity, 0.13 mIU/mL), <15.7%, <13.6% (sensitivity, 0.04 mIU/mL), and <3.7%, respectively. For DEHA-S, the inter- and intra-assay CVs were 4.83 % and 9.32% (sensitivity, 0.02 ng/mL), respectively, and for AMH, 14.2% and 12.3% (sensitivity of 0.14 ng/mL), respectively. Side effects related to the acupuncture treatment such as needling pain, hematoma, and numbness of legs were observed until the clinical trial was completed.

The primary outcome was the number of oocytes retrieved following IVF. Ovarian reserve was assessed at baseline for screening and before the induction of ovulation. Secondary outcomes were antral follicle count, Anti-Müllerian hormone level, and the number of fertilized oocytes.

### 2.5. Statistical Analyses

Statistical analyses were performed using SAS, version 9.4 (SAS Institute Inc., Cary, NC, USA). All analyses were based on the intention-to-treat principle. Categorical data are presented as numbers and percentages. For the comparison of continuous and categorical variables between groups, the Wilcoxon rank-sum test and chi-square test were used, respectively. Correlation and linear regression analyses between outcomes and maternal age were performed. *p* < 0.05 was considered statistically significant.

## 3. Results

### 3.1. Extraction of Candidate Acupoints for POR Treatment

A total of 110 records were identified in the database search, of which 81 records were excluded during initial screening. After reviewing the full texts of 29 clinical studies, 16 studies were excluded. Finally, 13 studies covering acupuncture (*n* = 4), electro-acupuncture (*n* = 5), and electrical stimulation of transcutaneous acupoints (*n* = 4) were included. The included data are summarized in Table 1. The PRISMA flow diagram is presented in Figure 3.

These 13 studies included the number of matured oocytes; rates of high-quality embryos; levels of fertility-related hormones such as follicle-stimulating hormone (FSH), luteinizing hormone (LH), and estradiol; and clinical rates of pregnancy as major outcomes. Among them, three studies showed beneficial effects of acupuncture on the number of matured oocytes (Table 1). The other six studies showed higher clinical pregnancy rates in the intervention group than in the control group. From these 13 reports, 27 acupoints were retrieved and included in the network analysis to determine the optimal acupoint combination for POR treatment (Table 2) [23,24,25,26,27,28,29,30,31,32,33,34,35].

### 3.2. Selection of Optimal Acupoint Combination for POR Treatment Using Network Analysis

The acupoint combination network extracted using the minimum spanning tree was split into three smaller groups. Acupoints such as CV4, SP6, and SP10 played the role of hub nodes in this network with a high degree of centrality and betweenness centrality (Figure 4). Based on the results of network analysis, three hub acupoints were selected. We selected five additional acupoints to increase or support the interaction of each acupoint through the Delphi method, which involved discussion with a Korean medicine expert group. Consequently, patients with POR were treated with acupuncture by needling at CV3, CV4, EX-CA1, LR3, KI3, SP6, SP10, and ST36.

### 3.3. Participant Characteristics

Thirty participants were initially screened, seven of whom were excluded owing to the incidence of adverse events (*n* = 0), failure in screening (*n* = 2), withdrawal of consent (*n* = 3), and infertility in the partner (*n* = 2). Twenty-three patients with POR were enrolled in this trial, but five of them could not participate owing to the inability to undergo acupuncture treatment for 2 months (*n* = 2), withdrawal of consent (*n* = 2), and loss to follow-up (*n* = 1). Figure 5 shows a flowchart of the patient enrollment protocol. The higher dropout rate in the Ac + IVF group than in the IVF group was attributed to a relatively long acupuncture treatment period of 2 months and a large number of visits (16 times).

### 3.4. Effectiveness of Acupuncture by Needling at Eight Acupoints Followed by IVF for POR Treatment

Table 3 shows the characteristics, including age, BMI, duration of infertility, number of COH cycles, basal hormone level, and AFC, of the 18 participants in the study. There were no differences in patient characteristics between the groups, except for serum LH levels. The basal LH level was higher in the Ac + IVF group than in the IVF group (*p* = 0.0155).

In the Ac + IVF group, the number of retrieved mature oocytes (2.75 ± 1.91) was slightly higher but not significantly more than that in the IVF group (1.30 ±1.25). Other secondary outcomes, including the Anti-Müllerian hormone level and antral follicle count, were similar between the groups. The number of zygotes (2.25 ± 1.39) was higher, but this was also not significantly different from that in the IVF group (1.10 ± 0.99) (Table 4). The rate of embryonic transfer in the Ac + IVF group was slightly higher, although not significantly more than that in the IVF group.

Subsequently, an analysis of women aged >37 years was conducted since accelerated oocyte loss begins at 37 years of age [36]. Interestingly, in these women, the number of retrieved mature oocytes and fertilized oocytes was significantly higher after acupuncture than after the IVF treatment alone without any other differences in patient characteristics (Table 5 and Table 6).

### 3.5. Inverse Correlation between the Number of Retrieved Mature Oocytes and COH Cycles Disappeared Following the Acupuncture Treatment

A decrease in the number of retrieved mature oocytes and fertilized oocytes associated with an increase in the number of COH cycles was observed in the IVF group (*p* < 0.05, Figure 6a). Interestingly, this inverse correlation was not observed when patients received acupuncture before ovulation induction. This finding was not influenced by the maternal age (Figure 6b). Therefore, a patient’s older age was unlikely to result in the retrieval of fewer mature oocytes for reasons other than the repeated IVF cycles.

We re-analyzed the data according to the number of consecutive COH cycles from one to five in both groups. In the Ac + IVF group, the number of retrieved mature oocytes was higher than that in the IVF group, with no differences in patients’ characteristics (Table 7 and Table 8). Further, we found that patients undergoing two to four COH cycles had significantly more retrieved mature oocytes after acupuncture treatment (Table 9, *p* < 0.05). There was also an increase in the number of fertilized oocytes after acupuncture treatment in patients who underwent two to three COH cycles (Table 9, *p* < 0.05).

## 4. Discussion

The impact of acupuncture treatment on infertility is still controversial. This is probably due to the various types of patients enrolled in randomized control trials and different prescriptions for acupoints leading to varying conclusions regarding the efficacy of acupuncture treatment [37,38]. Most randomized clinical trials included women with multifactorial infertility attributed to unexplained disorders or ovarian or uterine defects. Nevertheless, the stimulation of different acupoints could produce diverse results in patients affected by the same disease [39]. To overcome this limitation, we selectively enrolled patients diagnosed only with POR and constructed a complex network model analysis to select the appropriate therapeutic acupoints for those patients.

POR results in decreased oocyte production, cycle cancellation, and low pregnancy outcomes [40]. To date, many clinical and non-clinical studies have attempted to develop POR-specific ovulation protocols and to evaluate the effects of adjuvant therapy with hormone supplementation or acupuncture treatment [41]. Our study focused only on patients with POR, who showed a decrease in blood stromal flow [42]. A previous study reported that the aging-related decrease in the number of oocytes is linked to abnormal ovarian vascularization and decreased blood perfusion in the microenvironment of growing follicles [43]. Abnormal perifollicular vascularization and decreased ovarian stromal blood flow are associated with POR for the induction of ovulation during IVF treatment [44,45,46]. Acupuncture could increase the ovarian blood flow through the inhibition of ovarian sympathetic nerve activity [47,48]. A previous study reported that eight sessions of electro-acupuncture treatment over a period of 4 weeks alleviated the high uterine artery blood flow impedance of women with infertility [49]. In contrast, enhancement of ovarian vascularization with aspirin could improve ovarian responsiveness and pregnancy rates in patients undergoing IVF treatment [50], while L-arginine supplementation increases the vascularization of ovarian follicles, thus becoming more permeable to plasma proteins and stimulating circulation of FSH and the growth hormone in ovarian follicles [51]. The enhanced ovarian angiogenesis induced by acupuncture may supply various growth factors and lead to improved follicular development and ovulation in patients with infertility. Locations targeted by the acupuncture treatment were selected with the aim to restore fertility in POR patients by balancing the endocrine system and hormones. Based on our findings, we hypothesize that increased ovarian blood flow following acupuncture treatment could improve the response of patients with POR to ovulation induction.

Network model analysis is necessary for the development of new strategies to analyze the relevant characteristics of acupoints [52]. Through this model, we found three hub acupoints, namely, CV4, SP6, and SP10, which were mainly used to treat infertility in human and animal models [53,54,55]. Electro-acupuncture at CV4 in ovariectomized rats was reported to activate the Wnt signaling pathway, which is important for the regulation of ovarian folliculogenesis and control of steroidogenesis [53,56]. Acupuncture needling in CV4, SP6, and SP8 regulates the hormonal activity of FSH, LH, and GnRH to relieve dysmenorrhea in rats [54]. Electro-acupuncture at CV4, SP10, ST36, and other five acupoints improved the embryo quality and live birth rate in patients with polycystic ovarian syndromes [55]. As demonstrated in the literature review, the eight acupoints selected may have effects on hormonal regulation and pregnancy outcomes. Further, it was revealed that acupuncture treatment with more than five of the selected eight acupoints increased the number of mature oocytes or clinical pregnancy outcomes [21,24,29,30,33]. Advanced network analysis may provide optimal acupoint combinations by gaining an understanding of their physiological roles. Additionally, it may help propose standard treatment guidelines that could be the basis for the design of clinical trials.

We assessed the effect of acupuncture at eight acupoint combinations on oocytes yield in patients with POR. During the observation period, no severe side effects owing to acupuncture were reported in any group. Interestingly, acupuncture treatment significantly increased the number of retrieved mature oocytes and the fertilization rate in patients with POR who were aged > 37 years or in those who underwent more than one COH cycle (Table 5 and Table 7). We also observed an age-independent deterioration of the ovarian response across repeated ART cycles. Acupuncture may be appropriate for the treatment of patients with POR undergoing consecutive IVF cycles. Advanced age and multiple COH cycles are related to a worse prognosis [57]. Repeated ovarian stimulation influenced the defect rates of mature oocytes in mice and the rates of pregnancy and implantation in humans [58]. Follicles with hypertrophied granulosa cells may induce oocytes to experience an intrafollicular milieu with negative consequences for competence during growth [59]. However, electro-acupuncture stimulation at SP6, EX-CA1, LR3, and other three acupoints reduced granulosa cell death through the PI3K/Akt signaling pathway in rats with premature ovarian failure [60].

Our study has some limitations. The small number of enrolled patients limits the generalizability and implications of our findings. To overcome this, based on our study data, we are planning to conduct a subsequent clinical study including 23 patients in each group (level of significance = 5%, power = 80%, type of test = two-sided, dropout rate = 20%, sample size = 2 (z_α__/2_ + z_β_)^2^ σ^2^/(μ_c_ − μ_t_)^2^ : (μ_c_ − μ_t_ ) = 1.15, σ = 1.18). Further, a treatment period of two menstrual cycles seems to be too short to allow the activation and growth of early follicles into mature ones that are ready for ovulation in humans. A prolonged treatment period of >2 months could be considered to stimulate resting early primordial follicles in patients with diminished ovarian reserve. Furthermore, ovarian blood flow and ovarian volume measurements are necessary to evaluate the direct effects and mechanism of acupuncture treatment. Further studies examining the effect of acupuncture on clinical outcomes, including the pregnancy rate and live birth rate in patients with POR following acupuncture and IVF treatments, are required.

## 5. Conclusions

Acupuncture before IVF treatment has beneficial effects; it increases the number of retrieved mature oocytes or fertilized oocytes in patients with POR and aged > 37 years or undergoing more than one IVF cycle. These findings suggest that physicians can consider acupuncture for the treatment of women with POR and aged > 37 years or undergoing multiple IVF cycles.

## Figures and Tables

**Figure 1 jcm-10-02182-f001:**
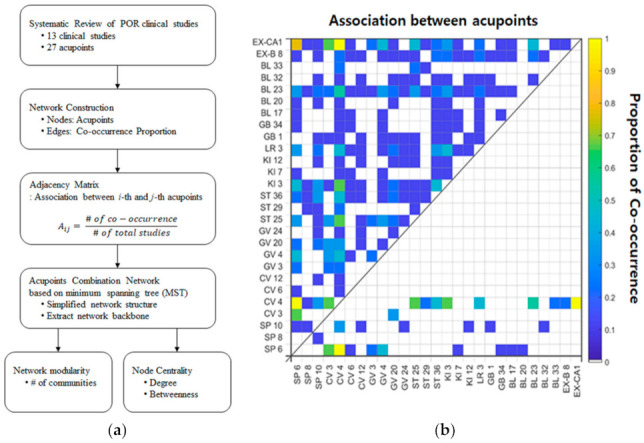
Network construction and adjacency matrix. (**a**) The procedure of network analysis involving the proportion of co-occurrence between acupoints revealed by previous clinical studies; (**b**) The association between acupoints is computed based on the proportion of co-occurrence between two acupoints, and the value is represented in the upper triangular part of the adjacency matrix. In the lower triangular part of the adjacency matrix, the value indicates the proportion of co-occurrence between two acupoints that survived as a result of the application of the minimum spanning tree algorithm. POR, poor ovarian response.

**Figure 2 jcm-10-02182-f002:**
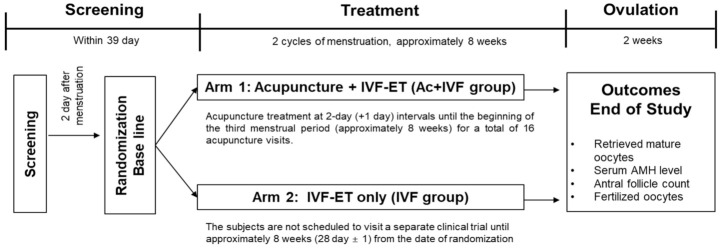
Flowchart of this trial. IVF, in vitro fertilization; AMH, Anti-Müllerian hormone.

**Figure 3 jcm-10-02182-f003:**
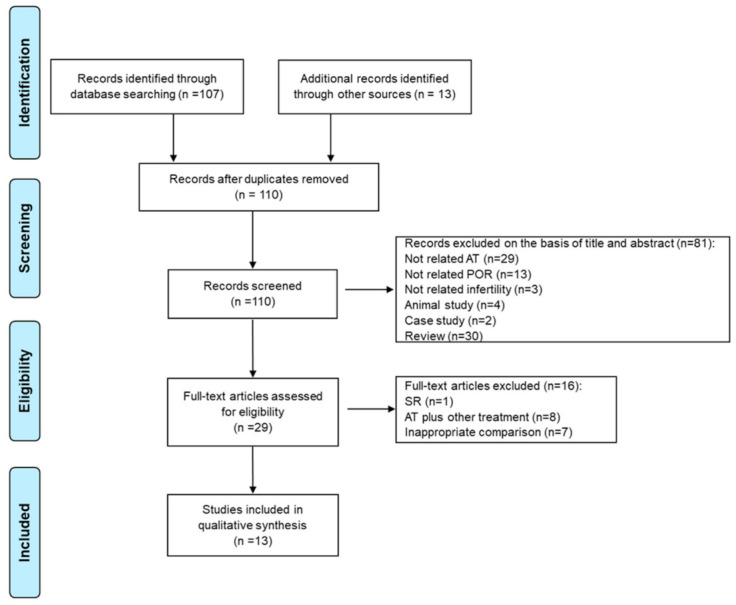
PRISMA flow diagram showing the extraction of candidate acupoints for the treatment of POR. PRISMA, preferred reporting items for systematic reviews and meta-analyses; AT, acupuncture treatment; POR, poor ovarian response; SR, systematic review.

**Figure 4 jcm-10-02182-f004:**
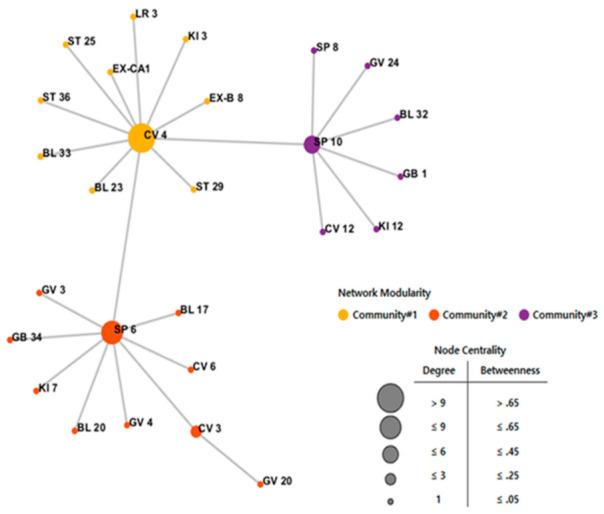
Acupoint combination network. The acupoint combination network simplified the network structure using the minimum spanning tree algorithm. The network consists of three communities centered on CV4, SP6, and SP10 acupoints. The size of each acupoint node was expressed as node centrality based on both degree and betweenness ranges.

**Figure 5 jcm-10-02182-f005:**
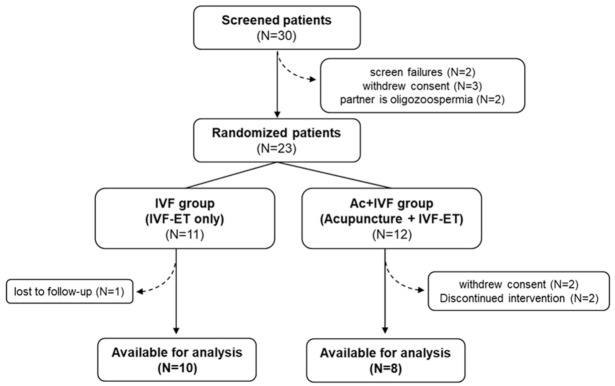
Flow chart of patient enrollment. Recruitment, follow-up, and dropouts over the course of study.

**Figure 6 jcm-10-02182-f006:**
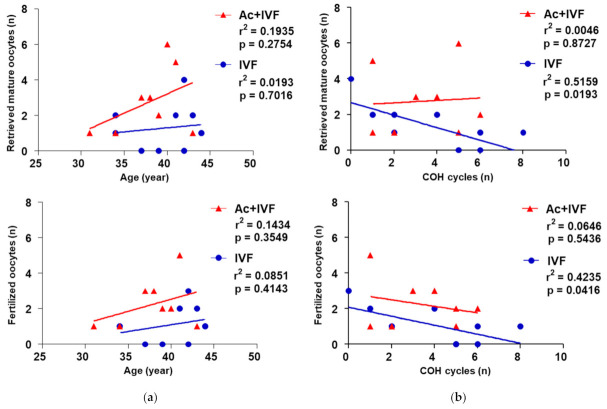
Comparison of correlation between clinical outcomes and the two characteristics in two study groups. Correlation of the number of retrieved mature oocytes and fertilized oocytes with the number of COH cycles (**a**) and age (**b**). Correlations are determined using multivariate linear regression analysis. IVF, in vitro fertilization; Ac + IVF, combination of acupuncture and IVF treatment; COH, controlled ovulation hyperstimulation.

**Table 1 jcm-10-02182-t001:** Summary of data from the systematic review.

First Author (Year)	Study Design; Sample Size; Age (y); Infertility Periods (y)	Intervention Group (A) Control Group (B, C, D)	Main Outcomes		Intergroup Differences	Acupoint
Zhou (2016) [23]	RCT; 63; (A)(35 ± 5)/(B)(36 ± 5); (A)(5.3 ± 4.6)/(B)(6.9 ± 5.2)	(A) AT plus IVF-ET (*n* = 30) (B) IVF-ET (*n* = 33)	(1) FSH (2) E2 (3) AFC (4) Number of matured oocytes (5) Fertilization rate	(6) High-quality embryo rate (7) Cycle cancellation rate (8) Implantation rate (9) Clinical pregnancy rate	(1–6) *p <* 0.05 (7–9) *p <* 0.01	SP6, CV4, CV6, GV4, ST36, KI3, KI7, LR3, GB34, BL17, BL23, EX-B8, EX-CA1
Xu (2016) [22]	RCT; 56; n.r.; n.r.	(A) AT (*n* = 28) (B) Estrogen and progesterone therapy (*n* = 28)	(1) FSH (2) AMH		(1–2) *p <* 0.05	SP6, SP10, CV4, GV20, ST36, KI3, LR3, BL20
Fernando (2016) [23]	Prospective observation study; 21; (A)(37.0 ± 5.4); (A)(3.2 ± 3.2)	(A) AT (*n* = 21)	(1) FSH (2) LH (3) E2	(4) FSH/LH ratios (5) Symptom scale scores	(1–5) *p <* 0.05	SP10, CV4, CV12, GV20, GV24, ST25, ST36, KI3, KI12, LR3, GB1, BL23, BL32, EX-B8
Yan (2015) [24]	RCT; 108; (29.8 ± 4.6); (4.9 ± 0.8);	(A) AT (*n* = 36) (B) Sham AT (*n* = 36) (C) No treatment (*n* = 36)	(1) Response rate (2) Planting rate	(3) Clinical pregnancy rate (4) Live birth rate	(1–4) *p <* 0.05	SP8, SP10, CV4, ST29, ST36, KI3, BL23, EX-CA1
Lian (2015) [25]	RCT; 66; (A)(38 ± 1)/(B)(37 ± 1); (A)(4.1 ± 2.2)/(B)(3.9 ± 1.9)	(A) EA plus IVF-ET (*n* = 33) (B) Sham EA plus IVF-ET (*n* = 33)	(1) Kidney deficiency syndrome score (2) High-quality embryo rate	3) Clinical pregnancy rate 4) IGF-1, IGF-2, β-EP	(1–2) *p <* 0.05 (3) *p >* 0.05 (5) *p <* 0.05	SP6, CV3, CV4, EX-CA1
Wang (2016) [26]	Prospective observation study; 21; (A)(37.0 ± 5.4); (A)(3.2 ± 3.2)	(A) EA (*n* = 21)	(1) FSH (2) LH (3) E2	4) FSH/LH ratios 5) Symptom scale scores	(1–5) *p <* 0.05	CV4, ST25, BL33, EX-CA1
Zhou (2013) [27]	Prospective observation study; 11; (A)25–39; (A) 0.33–10	(A) EA (*n* = 11)	(1) E2 (2) FSH (3) LH		(1) *p =* 0.002 (2) *p =* 0.001 (3) *p =* 0.002	CV4, ST25, ST29, BL33
Chen (2009) [28]	RCT; 60; (A)(34.3 ± 2.7)/(B)(34.6 ± 2.3); (A)(6.2 ± 4.3)/(B)(6.0 ± 4.6)	(A) EA plus IVF-ET (*n* = 30) (B) IVF-ET (*n* = 30)	(1) Kidney deficiency syndrome score (2) E2 (3) Fertilization rate (4) Number of matured oocytes	(5) High-quality embryo rate (6) Implantation rate (7) Pregnancy rate	(1–6) *p <* 0.05 (7) *p >* 0.05	SP6, CV4, KI3
Liu (2008) [29]	RCT; 60; (A)(34.3 ± 2.7)/(B)(34.6 ± 2.3); (A)(4.3 ± 1.4)/(B)(4.4 ± 1.3)	(A) EA plus GnRH antagonist (*n* = 30) (B) GnRH antagonist (*n* = 30)	(1) doses and days of GnRH (2) Thickness of endometrium (3) E2 (4) FSH (5) LH (6) SCF	(7) Fertilization rate (8) Number of matured oocytes (9) High-quality embryo rate (10) Pregnancy rate (11) Abortion rate	(1–2) *p >* 0.05 (3–11) *p <* 0.05	SP6, CV3, CV4, KI3, LR3, EX-CA1
Zheng (2015) [30]	RCT; 240; (A)(36.0 ± 5.4)/(B)(36.9 ± 4.3) (C)(36.8 ± 4.6)/(D)(36.8 ± 5.3); (A)(4.4 ± 2.9)/(B)(5.2 ± 3.2) (C)(4.7 ± 2.6)/(D)(4.8 ± 2.4);	(A) TEAS (*n* = 56) (B) No treatment (*n* = 60) (C) False Han’s placebo (*n* = 56) (D) Artificial endometrial cycle treatment (*n* = 54)	(1) E2 (2) FSH (3) AMH (4) LH (5) Number of matured oocytes (6) Number of embryos transferred	(7) Clinical pregnancy rate (8) Fertilization rate (9) Cleavage rate (10) High-quality embryo rate	(1–4) *p <* 0.01 (5–10) *p <* 0.05	SP6, CV3, CV4, GV4, GV20, ST25, BL23, EX-CA1
Lian (2014) [31]	RCT; 66; (A)(36 ± 2)/(B)(37 ± 3); (A)(4.4 ± 2.5)/(B)(4.3 ± 2.6)	(A) TEAS plus IVF-ET (*n* = 33) (B) Sham AT plus IVF-ET (*n* = 33)	(1) The kidney deficiency syndrome score (2) HCG E2/follicle count per day (3) High-quality follicle rate	(4) High-quality embryo rate (5) Clinical pregnancy rate	(1) *p <* 0.01 (2–3) *p <* 0.05 (4–5) *p >* 0.05	SP6, CV3, CV4, EX-CA1
Chen (2011) [32]	RCT; 80; (A)(37.1 ± 5.3)/(B)(36.6 ± 4.6); (A)(6.4 ± 4.4)/(B)(6.1 ± 3.0)	(A) TEAS plus IVF-ET (*n* = 40) (B) IVF-ET (*n* = 40)	(1) FSH (2) E2 (3) High-quality embryo rate	(4) Cryopreservation of embryo (5) Clinical pregnancy rate	(1–5) *p <* 0.05	SP6, CV3, CV4, GV3, GV4, ST25, EX-CA1,
Zhu (2012) [33]	RCT; 60; (A)(36.2 ± 1.0)/(B)(36.6 ± 0.8); (A)(4.4 ± 0.3)/(B)(4.5 ± 0.2)	(A) TEAS plus IVF-ET (*n* = 30) (B) IVF-ET (*n* = 30)	(1) Kidney symptom scores (2) FSH (3) E2 (4) Number of matured oocytes	(5) High-quality embryo rate (6) Cryopreservation of embryo (7) Clinical pregnancy rate	(1–7) *p <* 0.05	SP6, CV3, CV4, GV3, GV4, ST25, BL23, EX-CA1

AFC, antral follicle count; COH, controlled ovulation hyperstimulation; E2, estradiol; EA, electro-acupuncture; ESHRE, European Society of Human Reproduction and Embryology; FSH, follicle-stimulating hormone; GnRH, gonadotropin-releasing hormone; HANS, Han’s acupoint nerve stimulator; LH, luteinizing hormone; RCT, randomized clinical study; SCF, Stem cell factor; TEAS, Transcutaneous electrical acupoint stimulation; TEAS, Transcutaneous acupoint electrical stimulation.

**Table 2 jcm-10-02182-t002:** Commonly used acupoints for acupuncture treatment in patients with poor ovarian responses.

Acupoints		Acupuncture				Electro-Acupuncture					Transcutaneous Electrical Acupoint Stimulation			
		Zhou 2016 [23]	Xu 2016 [24]	Fernando 2016 [25]	Yan 2015 [26]	Lian 2015 [27]	Wang 2016 [28]	Zhou 2013 [29]	Chen 2009 [30]	Liu 2008 [31]	Zheng 2015 [32]	Lian 2014 [33]	Chen 2011 [34]	Zhu 2012 [35]
Sanyinjiao	SP 6	O	O			O			O	O	O	O	O	O
Diji	SP 8				O									
Xuehai	SP 10		O	O	O									
Zhongji	CV 3					O				O	O	O	O	O
Guanyuan	CV 4	O	O	O	O	O	O	O	O	O	O	O	O	O
Qihai	CV 6	O												
Zhongwan	CV 12			O										
Yaoyangguan	GV 3												O	O
Mingmen	GV 4	O									O		O	O
Baihui	GV 20		O	O							O			
Shenting	GV 24			O										
Tianshu	ST 25			O			O	O			O		O	O
Guilai	ST 29				O			O						
Zusanli	ST 36	O	O	O	O									
Taixi	KI 3	O	O	O	O				O	O				
Fuliu	KI 7	O												
Dahe	KI 12			O										
Taichong	LR 3	O	O	O						O				
Tongziliao	GB 1			O										
Yanglingquan	GB 34	O												
Geshu	BL 17	O												
Pishu	BL 20		O											
Shenshu	BL 23	O		O	O						O			O
Ciliao	BL 32			O										
Zhongliao	BL 33						O	O						
Shiqizhui	EX-B 8	O		O										
Zigong	EX-CA1	O			O	O	O			O	O	O	O	O

Twenty-seven acupoints from 13 clinical studies are included in the present pattern analysis.

**Table 3 jcm-10-02182-t003:** Clinical characteristics of patients enrolled in the two study groups.

Characteristic		IVF (*n* = 10)	Ac + IVF (*n* = 8)	*p*-Value
Age (y)		40.00 ± 3.83	37.88 ± 3.87	*p =* 0.2279
BMI (kg/m^2^)		23.14 ± 3.81	20.69 ± 2.99	*p =* 0.1728
Duration of infertility (months)		48.30 ± 36.59	33.25 ± 27.06	*p =* 0.5310
Number of COH cycles		3.90 ± 2.56	3.38 ± 1.92	*p =* 0.6530
Hormone (Basal level)	FSH (mIU/mL)	8.61 ± 8.25	17.62 ± 16.69	*p =* 0.2031
	LH (mIU/mL)	1.80 ± 0.95	7.74 ± 8.63	*p =* 0.0155
	E2 (pg/mL)	88.10 ± 69.62	120.86 ± 69.47	*p =* 0.2370
	TSH (µIU/mL)	2.70 ± 1.44	2.19 ± 0.89	*p =* 0.6334
	DHEAS (µg/dL)	139.87 ± 58.95	202.49 ± 144.60	*p =* 0.4082
	PRL (ng/mL)	10.34 ± 5.33	9.27 ± 5.07	*p =* 0.5726
	AMH (ng/mL)	0.44 ± 0.36	0.58 ± 0.44	*p =* 0.4232
Antral follicle count (Basal level)		3.80 ± 1.03	3.63 ± 0.92	*p =* 0.6033

Values are shown as mean ± standard deviation. Groups are compared using Wilcoxon rank-sum test. IVF, in vitro fertilization; Ac + IVF, combination of acupuncture and IVF treatment; COH, controlled ovulation hyperstimulation; FSH, follicle-stimulating hormone; LH, luteinizing hormone; E2, estradiol; TSH, thyroid-stimulating hormone; DHEAS, dehydroisoandrosterone sulfate; PRL, prolactin; AMH, Anti-Müllerian hormone.

**Table 4 jcm-10-02182-t004:** Comparison of clinical outcomes in the two study groups.

Outcome	IVF (*n* = 10)	Ac + IVF (*n* = 8)	*p*-Value
Number of retrieved mature oocytes	1.30 ± 1.25	2.75 ± 1.91	*p =* 0.0916
Number of fertilized oocytes	1.10 ± 0.99	2.25 ± 1.39	*p =* 0.0714
Fertilization rate * (No. of zygotes/No. of oocytes)	89.29 ± 19.67% (11/13)	91.67 ± 23.57% (18/22)	*p =* 0.8320
AMH (ng/mL) Antral follicle count	0.47 ± 0.43	0.55 ± 0.36	*p =* 0.4232
	5.60 ± 2.37	5.13 ± 1.89	*p =* 0.6033

Values are shown as mean ± standard deviation. Groups are compared using Wilcoxon rank-sum test. * calculated using the chi-square test. IVF, in vitro fertilization; Ac + IVF, combination of acupuncture and IVF treatment; AMH, Anti-Müllerian hormone.

**Table 5 jcm-10-02182-t005:** Clinical characteristics of patients aged > 37 years enrolled in the two study groups.

Characteristic		IVF (*n* = 10)	Ac + IVF (*n* = 8)	*p*-Value
Age (y)		40.67 ± 3.39	39.67 ± 2.16	*p* = 0.1724
BMI (kg/m^2^)		23.45 ± 3.90	21.89 ± 2.23	*p =* 0.2284
Duration of infertility (months)		46.67 ± 38.42	40.17 ± 27.99	*p =* 0.5511
Number of COH cycles		3.44 ± 2.24	3.33 ± 2.07	*p =* 0.7933
Hormone (Basal level)	FSH (mIU/mL)	9.35 ± 8.39	11.25 ± 11.54	*p =* 0.7982
	LH (mIU/mL)	1.83 ± 1.00	4.59 ± 3.27	*p =* 0.0537
	E2 (pg/mL)	70.83 ± 45.77	117.03 ± 64.12	*p =* 0.1419
	TSH (µIU/mL)	2.80 ± 1.49	2.00 ± 0.96	*p =* 0.3450
	DHEAS (µg/dL)	132.73 ± 57.76	222.75 ± 165.23	*p =* 0.1812
	PRL (ng/mL)	8.90 ± 2.96	8.29 ± 5.10	*p =* 0.4908
	AMH (ng/mL)	0.40 ± 0.37	0.59 ± 0.435.10	*p =* 0.3319
Antral follicle count (Basal level)		3.78 ± 1.09	3.67 ± 1.03	*p =* 0.9451

Values are shown as mean ± standard deviation. Groups are compared using Wilcoxon rank-sum test. IVF, in vitro fertilization; Ac + IVF, combination of acupuncture and IVF treatment; COH, controlled ovulation hyperstimulation; FSH, follicle-stimulating hormone; LH, luteinizing hormone; E2, estradiol; TSH, thyroid-stimulating hormone; DHEAS, dehydroisoandrosterone sulfate; PRL, prolactin; AMH, Anti-Müllerian hormone.

**Table 6 jcm-10-02182-t006:** Comparison of clinical outcomes of patients aged >37 years in the two study groups.

Outcome	IVF (*n* = 10)	Ac + IVF (*n* = 8)	*p*-Value
Number of retrieved mature oocytes	1.33 ± 1.32	3.33 ± 1.86	*p =* 0.0426
Number of fertilized oocytes	1.11 ± 1.05	2.67 ± 1.37	*p =* 0.0550
Fertilization rate * (No. of zygotes/No. of oocytes)	87.50 ± 20.92% (9/10)	88.89 ± 27.22% (16/20)	*p =* 0.4884
AMH (ng/mL) Antral follicle count	0.51 ± 0.44	0.45 ± 0.35	*p =* 0.8099
	5.44 ± 2.46	5.67 ± 1.86	*p =* 0.5542

Values are shown as mean ± standard deviation. Groups are compared using Wilcoxon rank-sum test. * calculated using the chi-square test. IVF, in vitro fertilization; Ac + IVF, combination of acupuncture and IVF treatment; AMH, Anti-Müllerian hormone.

**Table 7 jcm-10-02182-t007:** Clinical characteristics of patients enrolled in the two study groups undergoing more than one IVF cycle.

Characteristic		IVF (*n* = 10)	Ac + IVF (*n* = 8)	*p*-Value
Age (y)		40.43 ± 3.87	37.88 ± 3.87	*p =* 0.3096
BMI (kg/m^2^)		24.19 ± 4.16	20.69 ± 2.99	*p =* 0.1672
Duration of infertility (months)		43.43 ± 37.84	33.25 ± 27.06	*p =* 0.5305
Number of COH cycles		3.86 ± 2.12	3.38 ± 1.92	*p =* 0.3797
Hormone (Basal level)	FSH (mIU/mL)	10.45 ± 9.35	17.62 ± 16.69	*p =* 0.1911
	LH (mIU/mL)	1.68 ± 1.10	7.74 ± 8.63	*p =* 0.0542
	E2 (pg/mL)	54.17 ± 18.94	120.86 ± 69.47	*p =* 0.2359
	TSH (µIU/mL)	2.89 ± 1.53	2.19 ± 0.89	*p =* 0.8884
	DHEAS (µg/dL)	141.37 ± 63.23	202.49 ± 144.60	*p =* 0.5414
	PRL (ng/mL)	8.74 ± 3.38	9.27 ± 5.07	*p =* 0.6730
	AMH (ng/mL)	0.39 ± 0.40	0.58 ± 0.44	*p =* 0.2890
Antral follicle count (Basal level)		3.71 ± 1.25	3.63 ± 0.92	*p =* 0.6830

Values are shown as mean ± standard deviation. Groups are compared using Wilcoxon rank-sum test. IVF, in vitro fertilization; Ac + IVF, combination of acupuncture and IVF treatment; COH, controlled ovulation hyperstimulation; FSH, follicle-stimulating hormone; LH, luteinizing hormone; E2, estradiol; TSH, thyroid stimulating hormone; DHEAS, dehydroisoandrosterone sulfate; PRL, prolactin; AMH, Anti-Müllerian hormone.

**Table 8 jcm-10-02182-t008:** Comparison of clinical outcomes of patients aged > 37 years in the two study groups.

Outcome	IVF (*n* = 10)	Ac + IVF (*n* = 8)	*p*-value
Number of retrieved mature oocytes	1.00 ± 0.87	2.75 ± 1.91	*p =* 0.0300
Number of fertilized oocytes	0.71 ± 0.76	2.25 ± 1.39	*p =* 0.0305
Fertilization rate * (No. of zygotes/No. of oocytes)	87.50 ± 25.00% (8/9)	91.67 ± 23.57% (18/22)	*p =* 0.6271
AMH (ng/mL) Antral follicle count	0.31 ± 0.27	0.55 ± 0.36	*p =* 0.3537
	4.86 ± 2.34	5.67 ± 1.86	*p =* 0.9610

Values are shown as mean ± standard deviation. Groups are compared using Wilcoxon rank-sum test. * calculated using the chi-square test. IVF, in vitro fertilization; Ac + IVF, combination of acupuncture and IVF treatment; AMH, Anti-Müllerian hormone.

**Table 9 jcm-10-02182-t009:** Comparison of clinical outcomes depending on the number of consecutive IVF cycles in the two study groups.

Number of COH Cycles	Group	Number of Retrieved Mature Oocytes	*p*-Value	Number of Fertilized Oocytes	*p*-Value
≥1	IVF (*n* = 9)	1.00 ± 0.87	*p* = 0.0300	0.74 ± 0.76	*p* = 0.0305
	Ac + IVF (*n* = 8)	2.75 ± 1.91		2.25 ± 1.39	
≥2	IVF (*n* = 8)	0.88 ± 0.83	*p* = 0.0389	0.75 ± 0.71	*p* = 0.0252
	Ac + IVF (*n* = 6)	2.67 ± 1.86		2.00 ± 0.89	
≥3	IVF (*n* = 6)	0.67 ± 0.82	*p* = 0.0253	0.67 ± 0.82	*p* = 0.0308
	Ac + IVF (*n* = 5)	3.00 ± 1.87		2.20 ± 0.84	
≥4	IVF (*n* = 6)	0.67 ± 0.82	*p* = 0.0485	0.67 ± 0.82	*p* = 0.0599
	Ac + IVF (*n* = 4)	3.00 ± 2.16		2.20 ± 0.82	
≥5	IVF (*n* = 5)	0.40 ± 0.55	*p* = 0.0600	0.40 ± 0.55	*p* = 0.0600
	Ac + IVF (*n* = 3)	3.00 ± 2.65		1.67 ± 0.58	

The number of retrieved mature oocytes and fertilized oocytes are compared among patients undergoing consecutive IVF cycles from one to five. Values are shown as mean ± standard deviation. Groups are compared using Wilcoxon rank-sum test. IVF, in vitro fertilization; Ac + IVF, combination of acupuncture and IVF treatment; COH, controlled ovulation hyperstimulation.

## Data Availability

The datasets generated during the current study are available from the corresponding author on reasonable request.

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
