# Peer review of "Acupuncture for Poor Ovarian Response: A Randomized Controlled Trial"

_jcm, 2021, doi:10.3390/jcm10102182_

Round 1
Reviewer 1 Report
This is an interesting paper. It combining both systematic review and experimental (clinical trial).
The results described both the literature review and selected acupuncture method/location also the result from the clinical trial.
However, there was a missing link between the literature review and the experiment. The linkage between the literature review and the clinical trial was not reflected in the discussion and conclusion.
I suggest the author incorporate both literature review and results from clinical trial in both discussion and conclusion.
The conclusion should be expanded and more comprehensive.
regards,
Author Response
- Response: We performed a literature review of 13 studies to determine the candidate acupoints for POR treatment, and we mentioned this information in the Results section. We have discussed only the important outcomes obtained from our data in the Discussion section (lines 324-332).
- “As demonstrated in the literature review, the eight acupoints selected may have effects on hormonal regulation and pregnancy outcomes. Further, it was revealed that acupuncture treatment with more than five of the selected eight acupoints increased the number of mature oocytes or clinical pregnancy outcomes [21, 24, 29, 30, 33].”

Reviewer 2 Report
Novel and very nicely written manuscript.
The number of acupuncture sessions was very impressive.
Introduction (article line 38) recommend saying irregular menstrual "cycles" and (article line 67) instead of the word "assisted" consider saying "analyzed." Table 7, Age of Ac+IVF please edit the age as it says 397.88.
Thank you for your hard work and excellent contribution to the medical literature.
Author Response
- Response: We have revised the terms and corrected Table 7 as per your recommendation.

Reviewer 3 Report
- The review has been done well considering such heterogeneous data.
- Some typos require correction.
- End purpose of fertility treatment is pregnancy, not Number of retrieved /fertilized oocytes. Acknowledgement of such would be appropriate.
Author Response
1. The review has been done well considering such heterogeneous data.
- Response: Thank you for your positive assessment.
2. Some typos require correction.
- Response: We have checked for and corrected errors in our manuscript.
3. End purpose of fertility treatment is pregnancy, not number of retrieved/fertilized oocytes. Acknowledgment of such would be appropriate.
- Response: Among patients with POR, the oocyte yield leads to the cumulative birth rate (ref 11). The final outcome of our study was the number of retrieved and fertilized oocytes instead of pregnancy rates. However, the effect of acupuncture on pregnancy outcomes requires attention in future research (lines 358-360).

Reviewer 4 Report
The work presented for review is valuable, especially in terms of the literature review for the selection of optimal acupoint combination in similar studies. However, this overview is so extensive that the idea of the main aim of the work is lost. Perhaps a review article on this subject would be worthwhile? However, the proper study, while it is very well described and contains many valuable figures and tables, their value is very low due to the very small size of the research groups and amounts to n = 10 (IVF) and n = 8 (Ac + IVF). For comparison, the protocol presented in the latest work of Xu et al., 2020 (Effect of acupuncture on women with poor ovarian response: a study protocol for a multicenter randomized controlled trial) describing a very similar study included 70 women in each group, respectively. Such small research groups make statistical analysis and inference considerably more difficult.
In addition, the methodological part did not mention what analyzes were performed in the biological material and what methods.
Table 7 is a mistake in the age of women in the Ac + IVF group.
Author Response
The work presented for review is valuable, especially in terms of the literature review for the selection of optimal acupoint combination in similar studies. However, this overview is so extensive that the idea of the main aim of the work is lost. Perhaps a review article on this subject would be worthwhile? However, the proper study, while it is very well described and contains many valuable figures and tables, their value is very low due to the very small size of the research groups and amounts to n = 10 (IVF) and n = 8 (Ac + IVF). For comparison, the protocol presented in the latest work of Xu et al., 2020 (Effect of acupuncture on women with poor ovarian response: a study protocol for a multicenter randomized controlled trial) describing a very similar study included 70 women in each group, respectively. Such small research groups make statistical analysis and inference considerably more difficult.
- Response: Thank you for your pertinent comment. We conducted an exploratory clinical trial with a small sample size. Based on this study, we are planning to conduct a subsequent clinical study including 23 patients in each group. We have discussed the limitation of the small sample size in the Discussion section (lines 345-347).
“The small number of enrolled patients limits the generalizability and implications of our findings. To overcome this, based on our study data, we are planning to conduct a subsequent clinical study including 23 patients in each group (level of significance = 5%, power = 80%, type of test = two-sided, dropout rate = 20%, sample size = 2(zα∕2 + zβ)2 σ2 ∕ (μc - μt)2 : (μc - μt ) = 1.15, σ = 1.18).”
In addition, the methodological part did not mention what analyzes were performed in the biological material and what methods.
- Response: We have mentioned the methodology for the biological tests in the Methods section (lines 150-152).
“Clinical laboratory tests (i.e, hematology, blood chemistry, hormonal assay and urinalysis) were performed at the time of screening and after the acupuncture treatment in Pusan National University hospital by routine methods.”
Table 7 is a mistake in the age of women in the Ac + IVF group.
- Response: We have corrected the Table 7.

Round 2
Reviewer 4 Report
it is still unknown what analyzes were carried out in what biological material and with what methods. The notion of "routine methods" is not sufficient. The techniques used, their sensitivity and specificity should be reported.
Author Response
Response: Thank you for pointing this out. We have included the list of tests conducted in the study. 1) Hematology: Hemoglobin, hematocrit, RBC, WBC, differential WBC count, and platelet count Automatic Hematology Counter, XN-9100 and XN-3000 (Sysmex Korea Co., Ltd., Korea) 2) Blood chemistry: Alkaline phosphatase; BUN; creatinine; SGPT (ALT); SGOT (AST); albumin; total protein; total bilirubin; uric acid; glucose; and cholesterol, sodium, potassium, chlorine, calcium, and phosphorus levels Roche Modular DP (Tokyo, Japan) using an enzymatic colorimetric method 3) Urinalysis: Specific gravity; color; pH; and presence of protein, glucose, bilirubin, blood, and WBC Automatic Urinalysis Analyzer, UC-3500 and UF-5000 4) Endocrine function test: (1) FSH: Dream gamma-10 RIA (Shin Jin Medics Incs., Korea) - inter- and intra-assay coefficients of variation (CV) were < 5.72% each (sensitivity, 0.09 mIU/mL) (2) LH: Dream gamma-10 RIA (Shin Jin Medics Incs., Korea) - inter- and intra-assay CV were < 8.24% each (sensitivity, 0.13 mIU/mL) (3) E2: Dream gamma-10 RIA (Shin Jin Medics Incs., Korea) - inter- and intra-assay CV were < 15.7% each (4) TSH: Coat-A-Count TSH IRMA kit (SIEMENS, Ireland) - inter- and intra-assay CV were < 13.6% each (sensitivity, 0.04 mIU/mL) (5) DHEA-S: DHEA-Sulfate RIA (Beckman Coulter, USA) - inter- and intra-assay CV were 4.83% and 9.32%, respectively (sensitivity, 0.026 ng/mL) (6) PRL: an immunoradiometric assay (DIAsource Immunoassays S.A., Nivelles, Belgium) - inter- and intra-assay CV were < 3.7% each (7) AMH: AMH/MIC Enzyme Immunoassay (Beckman Coulter, France) - inter- and intra-assay CV were 14.2% and 12.3%, respectively (sensitivity, 0.14 ng/mL) We have included the methodology for the biological tests in the Methods section of the manuscript (Lines 151–160) “Clinical laboratory tests (i.e., hematology, blood chemistry, and urinalysis) and endocrine function tests were performed using routine methods at the time of screening and following acupuncture treatment at Pusan National University hospital. Serum follicle-stimulating hormone (FSH), luteinizing hormone (LH), estradiol (E2), and dehydroisoandrosterone sulfate (DHEA-S) were detected using a radioimmunoassay. Serum thyroid-stimulating hormone (TSH) and prolactin (PRL) were detected using an immunoradiometric assay. Serum anti-mullerian hormone (AMH) was detected using an enzyme immunoassay. For FSH, LH, E2, TSH, and PRL, the inter- and intra-assay coefficients of variations (CV) were < 5.72 % (sensitivity, 0.09 mIU/mL), < 8.24 % (sensitivity, 0.13 mIU/mL), < 15.7 %, < 13.6 % (sensitivity, 0.04 mIU/mL), and < 3.7 %, respectively. For DHEA-S, the inter- and intra-assay CVs were 4.83 % and 9.32 % (sensitivity, 0.02 ng/mL), respectively, and for AMH, 14.2 % and 12.3 % (sensitivity, 0.14 ng/mL), respectively.”